# Effects of *Sapindus mukorossi* Seed Oil on Skin Wound Healing: In Vivo and in Vitro Testing

**DOI:** 10.3390/ijms20102579

**Published:** 2019-05-26

**Authors:** Chang-Chih Chen, Chia-Jen Nien, Lih-Geeng Chen, Kuen-Yu Huang, Wei-Jen Chang, Haw-Ming Huang

**Affiliations:** 1Emergency Department, Mackay Momorial Hospital, Taipei 110, Taiwan; longus4280@gmail.com; 2Medical School, Mackay Medical College, New Taipei City 252, Taiwan; 3Graduate Institute of Biomedical Optomechatronics, College of Biomedical Engineering, Taipei Medical University, Taipei 110, Taiwan; yz26796029@hotmail.com (C.-J.N.); alvin199223@gmail.com (K.-Y.H.); 4Department of Microbiology, Immunology and Biopharmaceuticals, College of Life Sciences, National Chiayi University, Chiayi 600, Taiwan; lgchen@mail.ncyu.edu.tw; 5School of Dentistry, College of Oral Medicine, Taipei Medical University, Taipei 110, Taiwan; m8404006@tmu.edu.tw

**Keywords:** wound healing, *Sapindus mukorossi*, β-sitosterol, anti-inflammatory

## Abstract

*Sapindus mukorossi* seed oil is commonly used as a source for biodiesel fuel. Its phytochemical composition is similar to the extracted oil from *Sapindus trifoliatus* seeds, which exhibit beneficial effects for skin wound healing. Since *S. mukorossi* seed shows no cyanogenic property, it could be a potential candidate for the treatment of skin wounds. Thus, we evaluated the effectiveness of *S. mukorossi* seed oil in the treatment of skin wounds. We characterized and quantified the fatty acids and unsaponifiable fractions (including β-sitosterol and δ-tocopherol) contained in *S. mukorossi* seed-extracted oil by GC-MS and HPLC, respectively. Cell proliferation and migratory ability were evaluated by cell viability and scratch experiments using CCD-966SK cells treated with *S. mukorossi* oil. The anti-inflammatory effects of the oil were evaluated by measuring the nitric oxide (NO) production in lipopolysaccharide-treated RAW 264.7 cells. Antimicrobial activity tests were performed with *Propionibacterium acnes*, *Staphylococcus aureus*, and *Candida albicans* using a modified Japanese Industrial Standard procedure. Uniform artificial wounds were created on the dorsum of rats. The wounds were treated with a carboxymethyl cellulose (CMC)/hyaluronic acid (HA)/sodium alginate (SA) hydrogel for releasing the *S. mukorossi* seed oil. The wound sizes were measured photographically for 12 days and were compared to wounds covered with analogous membranes containing a saline solution. Our results showed that the *S. mukorossi* seed oil used in this study contains abundant monounsaturated fatty acids, β-sitosterol, and δ-tocopherol. In the in vitro tests, *S. mukorossi* seed oil prompted cell proliferation and migration capability. Additionally, the oil had significant anti-inflammatory and anti-microbial activities. In the in vivo animal experiments, *S. mukorossi* seed oil-treated wounds revealed acceleration of sequential skin wound healing events after two days of healing. The size of oil-treated wound decreased to half the size of the untreated control after eight days of healing. The results suggest that *S. mukorossi* seed oil could be a potential source for promoting skin wound healing.

## 1. Introduction

The soapnut tree, which belongs to the family Sapindaceae, is one of the most economically important trees found in tropical and subtropical climates from Japan to India in Asia [1,2,3]. Soapnut is known for its fruit, which contains triterpenoid saponins (10.1%) in the pericarp [3]. Saponin is a natural detergent for washing the body, hair, and clothes, and it is used as a natural surfactant [1,2,4,5,6]. There are more than 40 wild species in the genus *Sapindus* (family Sapindaceae) [6]. Among these species, *Sapindus mukorossi* (*S. mukorossi*) and *Sapindus trifoliatus* (*S. trifoliatus*) are the two main varieties. Recently, the use of saponins from *S. mukorossi* has gained the attention of investigators because of their various biological and pharmacological applications. It is reported that saponins exhibit properties that inhibit tumor cell growth [2]. These saponins also have anti-microbial activity [7] and reportedly can be used to treat eczema and psoriasis [3,8].

*S. mukorossi* is composed of about 56% pericarp [3], with the balance being the hard, black, smooth seed that contains the kernel inside [5,9] (Figure 1). The seed kernel of *S. mukorossi* contains 23% oil in the pulp, of which almost 90% are triglycerides [2]. Most of the reports of *S. mukorossi* are mainly about the nature and application of saponins in the pericarp part of the fruit. This situation is because the oil in the *S. mukorossi* seed kernel is inedible, and thus, the seed is usually treated as waste [1,10,11]. Recently, to achieve the goals of a “waste-to-energy” scheme, *S. mukorossi* seed oil was investigated as a potential source for the production of biodiesel fuel [10,12,13]. 

It is well known that many seed oils have therapeutic anti-inflammatory and antioxidant effects on the skin and promote wound healing and repair of skin [14,15,16]. In 2014, Pai et al. tested the n-hexane and ethyl acetate extracts of *S. trifoliatus* seeds and found that the seed extracts exhibited significant antibacterial, antifungal, and antioxidant activities, and revealed benefits for skin wound healing [17]. However, because *S. trifoliatus* seed extract contains cyanolipids, which is irritating or toxic to human skin [18], the pharmacological benefits are limited. Since the composition of *S. mukorossi* oil is very similar to that of *S. trifoliatus* oil [19] but without the toxic cyanolipids [20], it is reasonable to hypothesize that the oil extract of *S. mukorossi* seed kernel could provide similar pharmaceutical effects without the adverse effect of cyanolipid. 

The application of *S. mukorossi* (Wu Huan Zi in Chinese) seed kernel for antimicrobial and skin care is recorded in China’s traditional pharmaceutical book, Compendium of Materia Medica (Bencao Gangmu in Mandarin) some 500 years ago. In 2015, Srinivasarao et al. found that the extract of *S. mukorossi* exhibited antimicrobial and antioxidant activity and suggested that *S. mukorossi* could be used as a potential source of natural antimicrobial treatment because it possessed strong antioxidant potential [21]. However, in their report, the extract was from whole *S. mukorossi* fruit; thus, it is difficult to know whether the antimicrobial and antioxidant effects were from the pericarp or the seed kernel. Recent studies indicate that *S. mukorossi* oil has abundant phytosterols and arachidonic acid (23.85%) [1,19]. It is reported that phytosterols have an anti-inflammatory effect on skin and are capable of reducing swelling and erythema [22]. Additionally, arachidonic acid is not only an essential polyunsaturated fatty acid in the skin [19] but also plays an important role in reducing skin inflammation [23]. Since *S. mukorossi* seed extract contains anti-inflammatory, antimicrobial, and antioxidant compounds, it could serve as a potential treatment for skin and soft-tissue infections (SSTIs) [24]. However, the direct topical application of *S. mukorossi* seed oil on the skin has not yet been well investigated. Accordingly, the purpose of this study was to test the pharmacological effects of *S. mukorossi* seed oil on skin wound healing in both in vivo and in vitro experiments.

## 2. Results

### 2.1. GC-MS Analysis 

The fatty acid composition of the *S. mukorossi* seed-extracted oil was determined by gas chromatography-mass spectrometry (GC-MS) analysis. A representative chromatogram is shown in Figure 2a. The identity of the peak was analyzed using fatty acid standards and the MS database. The percentages of fatty acid esters were obtained by calculating the peak area ratios, which are listed in Table 1. The results show that *S. mukorossi* seeds contain 5.35% of palmitic acid (C16:0), 0.9% stearic acid (C18:0), 52.46% oleic acid (C18:1), 7.19% linoleic acid (C18:2), 1.61% linolenic acid (C18:3), 6.84% arachidic acid (C20:0), 23.71% eicosenic acid (C20:1), 1.24% bechenic acid, and 0.68% erucic acid (C22:1). 

### 2.2. HPLC Analysis

High-pressure liquid chromatography (HPLC) was performed to determine the quantities of δ-tocopherol and β-sitosterol. Table 2 provides the linear calibration curves for the standard solutions of the two analytes. As shown in Figure 2b,c, the retention times for δ-tocopherol and β-sitosterol were 12.67 min and 43.40 min, respectively. The total amount of δ-tocopherol and β-sitosterol in the *S. mukorossi* seed oil was 73.9 ± 23.6 μg/mL and 232.64 ± 4.5 μg/mL, respectively.

### 2.3. Antimicrobial Activity Testing

The antimicrobial activity of *S. mukorossi* seed oil is shown in Table 3. After treatment with 1% *S. mukorossi* seed oil for 24 h, the inhibition rates for *Propionibacterium acnes*, *Staphylococcus aureus*, and *Candida albicans* reached 99%. 

### 2.4. Anti-Inflammatory Testing 

As shown in Figure 3, *S. mukorossi* seed oil significantly affected nitric oxide (NO) release in the lipopolysaccharide (LPS)-treated cells. The NO release of the samples was normalized by comparing the measured data to the untreated samples. When the cells were treated with *N*-nitro-l-arginine-methyl ester (l-NAME), 21% NO release was noted compared to the control. When the cells were pretreated with 25 μg/mL *S. mukorossi* seed oil, NO release by the lipopolysaccharide (LPS)-treated RAW 264.7 cells was 92%. This value dramatically decreased to 46% when 500 μg/mL oil was used for pretreatment.

### 2.5. Cell Proliferation Assay

The in vitro cell experiments demonstrated that the cells treated with *S. mukorossi* oil exhibited a traditional growth curve. The seed oil-extract significantly enhanced the viability of the tested CCD-966SK cells on day 2 (Figure 4) (*p* < 0.05). After three days of culture, *S. mukorossi* seed oil showed no proliferation or cytotoxicity effect on the cultured cells. The scratch assay results showed that treatment with *S. mukorossi* seed oil enhanced the CCD-966SK cell migration toward the scratched area (Figure 5). The microscopic images revealed that the leading cells at the wound edge oriented towards the wound area 6 h after the scratch trauma was inflicted (Figure 5b,f). Additionally, the quantitative analysis of wound closure (Figure 6) showed that wound closure rate for the oil-treated cells was 24.73%, which is higher than that of the control cells (6.45%) at 6 h. After a 12-hour culture period, the border of the wound became unclear. In addition, a greater number of migrating cells were noted at the scratch edge in the oil-treated sample than in the control group (Figure 5c,g). The oil-treated cells demonstrated a 3.36-fold higher percentage of scratch-width closure than the control cells. At 24 h, the migrating cells had moved to the center of the scratch wound (Figure 5d,h). The oil-treated CCD-966SK cells (Figure 5h) displayed significantly increased healing ability compared to the controls (Figure 5d). The wound width closure of the oil-treated group after 24 h of culture was 82.79%, which was 1.88-fold greater than that of the controls (44.08%) (Figure 6). 

### 2.6. In Vivo Wound Healing Experiment

The in vivo skin wound healing activity of *S. mukorossi* seed oil extract is shown in Figure 7. Two days after the skin excision, the typical healing responses inside the epidermis of the wound were obviously better in the group treated with *S. mukorossi* seed oil. During the initial two days of healing, the wounds of both the oil-treated and control groups developed hyperemic areas with well-defined borders that preserved the rectangular shape of the wound. In the oil-treated group, the wounds underwent accelerated healing with the growth of granulation tissue, absence of edema, and lower secretions than in the untreated control group. In addition, a quantitative assessment demonstrated that the in vivo experimental wounds treated with *S. mukorossi* seed oil (74.14 ± 1.64%) showed a statistically significant reduction (*p* < 0.05) in the wound area compared to that of the control wounds (91.02% ± 7.44%) (Figure 8). This statistically significant reduction in wound size was observed at all the experimental time points. 

On day 8, the wounds had lost their geometrical shape whether or not they had been treated with *S. mukorossi* seed oil. However, compared to the control group, rats treated with *S. mukorossi* seed oil demonstrated no secretion in the wound bed. In addition, the wound became dark brown, dry, and smaller than those in the control group. The wound size of the oil-treated rats decreased significantly to 25.30% ± 6.98% (Figure 8), which is almost half that of the untreated control wounds (42.45% ± 7.95%) (*p* < 0.05). 

On the 12^th^ day, the untreated wounds also demonstrated wound beds without secretions and hyperemia (Figure 7). The brown color and dryness of the untreated wounds on day 12 were similar to the oil-treated wounds on the 8^th^ day. However, the treated wounds showed complete resurfacing of the epithelial layer, and the wound was almost entirely covered by hair. At this healing stage, the average percentage wound size for the control and oil-treated wounds was 13.55% ± 3.40% and 7.20% ± 1.52%, respectively (Figure 8). 

## 3. Discussion

Wound healing is a series of processes that involves control of inflammation, proliferation, and new tissue remodeling [16,25,26]. Among these processes, inflammation is the first step in the healing response after tissue injury. In addition, cell proliferation and migration are essential responses for re-epithelialization and skin remodeling during the healing process [26]. Our results showed an improvement in CCD-966SK cell proliferation (Figure 4) and migration (Figure 5 and Figure 6) induced by *S. mukorossi* seed oil. Similar healing effects were also observed in the artificial wound in the rat model (Figure 7 and Figure 8). These phenomena are similar to those reported by de Moura Sperotto et al. who tested the wound healing effects of a *Plantago australis* extract and concluded that such plant extracts have an action at the end of the proliferative phase or in maturation phase that promotes the normalization of the tissue [26]. 

The phytochemical characteristics of the *S. mukorossi* seed oil used in this study were determined by a series of tests. The GC-MS results show that the fatty acid contained in the *S. mukorossi* seeds oil was similar to that of previous reports [9,19,20]. The most significant phytochemical finding of the tested *S. mukorossi* seed oil is that it contains a substantial amount of unsaturated fatty acids, which account for 85.65% of all the fatty acids in the seed oil. Among these unsaturated fatty acids, 76.85% were monounsaturated fatty acids. It is reported that monounsaturated fatty acids may cover the skin barrier and act as permeability enhancers [16]. Accordingly, these fatty acids have been wildly used not only in cosmetology for daily care of the face and body but also in the acceleration of skin wound healing [27].

Interestingly, Table 1 shows that the main components of the monounsaturated fatty acids in *S. mukorossi* seed oil were oleic acid (52.46%) and eicosenic acid (23.71%). Since oleic acid strongly inhibited the production of NO at the wound site [16] and were positive for wounding healing, it is not a surprise that the tested *S. mukorossi* seed oil strongly promoted the skin wound healing process as shown in Figure 7 and Figure 8. Except for monounsaturated FAs, linoleic acid also plays a role in maintaining the integrity of the skin barrier for maintaining water permeability. In Table 1, we show that the extracted *S. mukorossi* seed oil contains 7.2% linoleic acid and 23.71% eicosenic acid. Since both linoleic acid and eicosenic acid directly play a role as activators that enhance skin cell proliferation [16,27], they may also play a role in CCD-966SK cell proliferation and migration as shown in Figure 4 and Figure 5.

The results of in vitro cell migration (Figure 5 and Figure 6) and in vivo animal skin wound healing (Figure 7 and Figure 8) showed that *S. mukorossi* seed oil enhanced skin wound healing. In 2018, Lin et al. reviewed the pharmacological and medical effects of several plant oils and concluded that the therapeutic benefits for skin wound healing of plant oils are provided by their antibacterial, anti-inflammatory, and antioxidant effects. We found that *S. mukorossi* seed oil is rich in β-sitosterol and δ-tocopherol (vitamin E) (Figure 2b,c). The total amount of β-sitosterol and δ-tocopherol in the *S. mukorossi* seed oil are 7.3% and 0.235%, respectively. These values are much higher than the analogous quantities in shea butter (0.8% for sterols and 0.08% for tocopherols) [28]. Since Shea butter is a famous anti-inflammatory and antioxidant plant seed extract used in the cosmetic industry because of its high percentage of unsaponifiable compounds (including phytosterols and tocopherol) [16,29], *S. mukorossi* seed oil could also have potential use in skin care. 

β-Sitosterol is the major phytosterol in plant oils. It provides a biological function like cholesterol and provides pharmacological and biological activities useful for the treatment of various skin illnesses, such as swelling and erythema [22,30] because of its structural similarity to cholesterol [31,32]. It was reported as a safe chemical without undesirable side effects [32,33]. Several studies tested the effects of phytosterols as anti-inflammatory, angiogenic, and cell migration stimulators and have confirmed its positive effects on skin barrier recovery and skin wound repair [14,16,31,32,34,35]. Since the *S. mukorossi* seed oil contains abundant β-sitosterol, it also showed the potential to be a skin wound healing enhancer.

Bioactive plant oil extracted from fruit pulp always contains not only β-sitosterol but also vitamin E [16,27]. The primary role of vitamin E in plants is to provide antioxidant activity [16]. It is well known that the production of reactive oxygen species (ROS) during skin injuries inhibits the healing process by various biological mechanisms [14,36]. The topical use of vitamin E is to attenuate oxidative stress by inhibiting the production of oxidase and NO.

In this study, δ-tocopherol was also detected in considerable amounts in the extracted *S. mukorossi* seed oil extracts. Thus, in Figure 4, the addition of *S. mukorossi* seed oil significantly decreased the NO release in LPS-treated RAW 264.7 macrophage cells in a dose-dependent manner. Since β-sitosterol also stimulates antioxidant enzymes and plays a role in ROS scavenging, it is not surprising that *S. mukorossi* seed oil exhibits promising beneficial effects and improved skin wound healing (Figure 7 and Figure 8). Rekik et al. also reported that the beneficial effect on skin wound healing and collagen synthesis of vitamin E and phytosterols is because these compounds prevent the damaging effects of free radicals and ensure the stability and integrity of biological membranes [14]. 

The mechanism of the antimicrobial activity of S. *mukorossi* seed oil shown in Table 3 is not fully understood. The most likely mechanism of the antimicrobial action of S. *mukorossi* seed oil is the β-sitosterol content because it is a potent antimicrobial agent at low concentrations [33]. In addition, it is reported that acidic plant oils are not conducive to the growth of bacteria, which require a neutral pH environment for growth [14,32]. Such an acidic environment also leads to the promotion of cell proliferation (Figure 4) and cell migration (Figure 5 and Figure 6). These effects construct an ideal environment for fibroblast activity and collagen reorganization, with resulting acceleration of wound healing [14,37]. Since fibroblast proliferation and migration are vital steps in skin wound healing [12] when this phenomenon is considered in combination with the efficiency of skin wound healing shown in Figure 7 and Figure 8, we suggest that S. *mukorossi* seed oil is a candidate agent for development as a skin treatment agent. 

## 4. Materials and Methods

### 4.1. Chemicals and Reagents

The reference standards of the fatty acids β-sitosterol, δ-tocopherol, lipopolysaccharide (LPS), CDCl_3_, Griess reagent (N5751) and *N*-nitro-l-arginine-methyl ester (l-NAME) were purchased from Sigma-Aldrich (St. Louis, MO, USA). Methanol, tetrahydrofuran acetonitrile, sodium hydroxide, and boron trifluoride/methanol were obtained from Fisher Scientific (Pittsburgh, PA, USA). DMEM (Dulbecco’s modified Eagle medium), FBS (fetal bovine serum), trypsin-EDTA, l-glutamine, penicillin/streptomycin were obtained from HyClone (South Logan, UT, USA). The tetrazolium salt (MTT) kit was purchased from Roche Applied Science (Mannheim, Germany). Isoflurane was purchased from DS Pharma Animal Health Co. (Osaka, Japan).

### 4.2. Plant Material

The *S. mukorossi* seeds were purchased from He He Co. Ltd. (Taipei, Taiwan). Before the oil extraction, the seeds were cleaned under running tap water followed by rinsing with sterile distilled water and then dried in an oven at 40 °C for 72 h. The seeds were then crushed using a grinder, and the kernels were separated from their hard shells. As previously reported, the oil was extracted using a cold press method and was followed by filtering (0.45 μm pore size) [19]. The oil recovery rate from the kernels was about 30%.

### 4.3. Phytochemical Analysis of Kernel Oil

#### 4.3.1. GC-MS Analysis

For the GC-MS analysis, the *S. mukorossi* seed-extracted oil was transesterified to produce fatty acid methyl esters (FAME). Briefly, 20 mg of extracted oil was mixed with 1 mL of 1 N sodium hydroxide to act as a catalyst. The mixture was then stirred vigorously using a magnetic stirrer at room temperature for 30 s. Then, for saponification, the mix was maintained at 80 °C for 15 min. Then, 1 mL of boron trifluoride in a methanol solution was added to the sample, and the mixture was shaken for 30 s. The mixture was then placed in a 110 °C dry bath for 15 min. After the sample had returned to room temperature, 1 mL of n-hexane was added to the solution. The polar layer was separated and used to inject into the GC-MS system (GCMS-QP2010, Shimadzu, Tokyo, Japan).

A BPX70 capillary column with a dimension of 30 m × 0.25 mm i.d. (0.25 μm film thickness) was used for oil composition separation. Helium was used as carrier gas at a pressure of 75 kPa. The sample was injected at a temperature set at 250 °C. The programmed temperature of the oven started at 120 °C and was held for 0.5 min for solvent delay, then increased at a rate of 10 °C/min to 180 °C, and then increased at a rate of 3 °C/min to 220 °C, followed by an increase of 30 °C/min to 260 °C and then, held for 5 min. The mass spectrometer was operated in the electron impact (EI) mode at 70 eV. The temperature of the ionization chamber was set at 200 °C. For identification of the analyzed constituents with various retention times, a library search of mass spectra was performed using NIST/EPA/NIH Mass Spectral Library.

#### 4.3.2. HPLC Analysis

HPLC was performed to determine the content of β-sitosterol and δ-tocopherol in the *S. mukorossi* seed oil. The HPLC was equipped with a low-pressure mixing pump (L2130, Hitachi, Tokyo, Japan), controlled by a CBM-20A interface module (Shimadzu Technology, Kyoto, Japan), and had a UV detector (Waters 486, Waters Corporation, Milford, MA, US). The separation was achieved using a Mightysil RP-18 GP (250 mm × 4.6 mm i.d., Mightysil RP 18 GP Cica, Tokyo, Japan) at 30 °C. For the β-sitosterol analysis, the mobile phase consisted of 96% methanol, 3% tetrahydrofuran, and 1% deionized water. For δ-tocopherol detection, pure methanol was used as the mobile phase. For testing the two molecules, 20 μL of *S. mukorossi* seed oil were injected. The flow rates for the detection of β-sitosterol and δ-tocopherol were 0.5 mL/min and 0.5 mL/min, respectively. All the samples were monitored at a wavelength of 280 nm. The compounds were identified by comparing their retention times to those of authentic standards. The quantification was achieved using linear regression analysis.

### 4.4. Antimicrobial Assay

The antimicrobial effects of *S. mukorossi* seed oil were tested in this study. The procedure was modified from the Japanese Industrial Standard JIS Z 2801:2000. Several microorganisms (Bioresource Collection and Research Center, BCRC, Hsinchu, Taiwan) including two bacteria (*P. acnes*; ATCC 11827 and *S. aureus*; ATCC 6538P) and a fungus (*C. albicans*; ATCC 10231) that cause skin diseases. All selected microbes were cultured in their standard culture medium. For testing the antimicrobial activity of the oil, 0.1 mL oil was mixed and inoculated with 10 mL of each microorganism culture to a concentration of 10^7^ colony-forming units (CFU)/mL at 25 °C for 24 h. After serial dilution, the three bacteria were swabbed uniformly across the agar surface of Petri plates. After 48 h incubation at 23 °C for *C. albicans* or 37 °C for the bacteria, the number of colonies on each plate was counted.
Inactivation rate (%) = [1 – (CFU_sample_/CFU_control_)] × 100(1)
where CFU_sample_ and CFU_control_ are the colony numbers of the tested and control samples, respectively. 

### 4.5. Anti-Inflammatory Test

To test the in vitro anti-inflammatory activity of *S. mukorossi* seed oil, we assessed the oil in LPS-induced RAW 264.7 cells. RAW 264.7 macrophage cells were seeded into a 96-well plate at a concentration of 4 × 10^5^ cells/mL. The cells were maintained in Dulbecco’s modified Eagle medium (DMEM) supplemented with 10% fetal bovine serum (FBS), and 1% penicillin/streptomycin and were cultured in an incubator at 37 °C and 5% CO_2_. After pre-incubation of the cells for 24 h, cells were pretreated with the *S. mukorossi* seed oil with a series of concentrations ranging from 25 to 500 μg/mL for 1 h and were further stimulated with LPS (1 μg/mL) from *Escherichia coli* strain 055:B5 for 24 h. *N*-nitro-l-arginine-methyl ester (l-NAME) at a concentration 1 mM was used as a positive control. The NO concentration produced by the RAW 264.7 cells was determined through the Griess assay. Briefly, an equal volume of Griess reagent was mixed with the culture supernatant, and color development was measured at 530 nm using a microplate reader (EZ Read 400, Biochrom, Holliston, MA, USA). Anti-inflammatory activity was presented in terms of NO production percentage. 

### 4.6. In Vitro Skin Cell Analysis

#### 4.6.1. Cell Proliferation Assay

For testing the proliferation effect of the *S. mukorossi* seed oil on skin cells, a cell viability assay was performed according to a previous study [36]. The human skin fibroblast cell line CCD-966SK (ATCC CRL-1881) was used for this in vitro cell analysis. The cells were seeded in 24-well plates at a concentration of 2 × 10^4^ cells/mL and were maintained in DMEM supplemented with 10% FBS and 1% penicillin/streptomycin. The cells were then incubated in an environment of 5% CO_2_ at 37 °C and 100% humidity. The viability of the CCD-966SK cells exposed to *S. mukorossi* seed oil at a concentration of 200 μg/mL was evaluated after three days of incubation. The cell viability was assessed using the tetrazolium salt (MTT) method. After the cells were incubated with the tetrazolium salt for 4 h, 500 μL of DMSO were added to solubilize the formazan dye overnight. Since no toxic effects were observed on the cells, the emulsifier polyoxyethylene sorbitan mono-oleate (Tween 80) was employed as the delivery vehicle [38]. The optical density was determined using a microplate reader (EZ Read 400, Biochrom, Holliston, MA, USA) at 570 and 690 nm.

#### 4.6.2. Scratch Wound Healing Test

The effect of *S. mukorossi* seed oil on cell migration behavior and wounded healing activity was assessed using scratch tests. Before the test, 2 × 10^4^ cells/mL CCD-966SK cells were cultured in 6-well Petri dishes and incubated in 5% CO_2_ at 37 °C and 100% humidity. When the cells reached confluence, they were starved overnight. A 1000-μL pipette tip was used to create a wound across the center of the culture dish. The scratched cells were washed with fresh medium to remove any loose or dead cells followed by exposure to 25 μg/mL *S. mukorossi* seed oil. Cells cultured with fresh medium but without oil served as a control group. For both groups, cell migration photographs were taken at 4× magnification using a bright-field illumination microscope (Eclipse TS100, Nikon Corporation, Tokyo, Japan). The images were captured by a digital camera (SPOT Idea, Diagnostic Instruments, Inc., Sterling Heights, MI, USA). The images were taken immediately (0 h) and after 6 and 12 h at five different sites from each wound area (gap). The captured images were analyzed using ImageJ software (National Institutes of Health, Bethesda, MD, USA) to evaluate the percentage of scratch closure, the percentage of gap closure was measured and compared with the results obtained before treatment (day 0) using the following formula [39]: [(A_0h_ − A_Δh_)/A_0h_] × 100%(2)
where A_0h_ is the scratch area at the beginning of the culture period, and A_Δh_ is the scratch area after a certain culture period. The increase in the percentage of the closed area reflects the migration of cells.

### 4.7. Wound Healing Activity Test

Healthy male Sprague-Dawley (SD) rats weighing 210–290 g were used to assess the effects of *S. mukorossi* seed oil on skin wound healing. The rats were obtained from the Laboratory Animal Center at the National Applied Research Laboratories (Hsinchu, Taiwan). The animals were kept in hygienic cages during the experimental period. The environment was maintained with a 12-hour light/dark cycle, a temperature at 21 °C, and a humidity of 60% to 70%. The study protocol and procedures were reviewed and approved by the Institution Animal Care and Use Committee or Panel (IACUC Approval No. L10708, 16 October 2018, and all efforts were made to minimize animal number and suffering to produce reliable scientific data. 

The wound healing experimental procedure was performed according to a previous study [10,40]. A carboxymethyl cellulose (CMC)/hyaluronic acid (HA)/sodium alginate (SA) hydrogel [41,42] was prepared on one side of a non-woven fabric for releasing the extracted *S. mukorossi* seed oil. The ratio of the CMC, HA, and SA in the hydrogels was 1:3:12. Before the study, the rats’ backs were one-way clipped (5 × 5 cm) with an electric animal shaver and the disinfected with 75% alcohol. Then, the rats were anesthetized with 5% isoflurane in an induction chamber. Before the experiment, one linear wound with an area of 2 × 2 cm was made by excising the skin on the back of the rat using sterile scissors.

Eight SD rats were randomly divided into two groups. For the tested group, the wound sites of the rats were covered with the prepared oil-hydrogel and were wrapped with porous bandages. The oil-free hydrogel was applied to the wounds of the control animals. The covered hydrogel was replaced every two days during the 12-day experiment. The rats were housed individually after wounding. For the evaluation of the progressive change in the wounded area, the wounds were photographed every two days with a digital camera. The wound area was measured using ImageJ software (National Institutes of Health, Bethesda, MD, USA). The wound area was determined by measuring the mass of the transparent paper cut to the shape of the wound. The wound contraction was expressed as a percentage reduction of the original cut size. The percentages of wound size were determined by calculating the ratio between the measured wound area and the original wound area.

### 4.8. Statistical Analysis

The results are presented as mean ± standard deviation (SD), and the comparison between groups was analyzed using the Student *t*-test. A *p* value less than 0.05 was considered statistically significant.

## 5. Conclusions

Although various medicinal plant oils are used to treat different kinds of wounds, the scientific evidence about whether *S. mukorossi* seed oil also provides benefits for skin wound healing is incomplete. The present investigation, for the first time, indicates that *S. mukorossi* seed oil extract shows remarkable antibacterial, anti-inflammatory, antioxidant, cell proliferation, cell migration stimulation, and skin wound healing. The total amount of δ-tocopherol and β-sitosterol in the *S. mukorossi* seed oil was 73.9 ± 23.6 μg/mL and 232.64 ± 4.5 μg/mL, respectively. The inhibition effect of 1% *S. mukorossi* seed oil on *P. acnes*, *S. aureus*, and *C. albicans* was 99%. Addition of 500 μg/mL *S. mukorossi* seed oil resulted in a reduction of NO release by the lipopolysaccharide (LPS)-treated RAW 264.7 cells to 46%. The wound size of the oil-treated rats decreased significantly to almost 50% when *S. mukorossi* seed oil rich membrane was used as a dressing material. In accordance with our results, we suggest that *S. mukorossi* seed oil has the potential for development to promote skin wound healing. Since the use of *S. mukorossi* seed oil involves a strategy of using a waste as source of bioactive compounds, it can conduct a cost benefit for skin care applications.

## Figures and Tables

**Figure 1 ijms-20-02579-f001:**
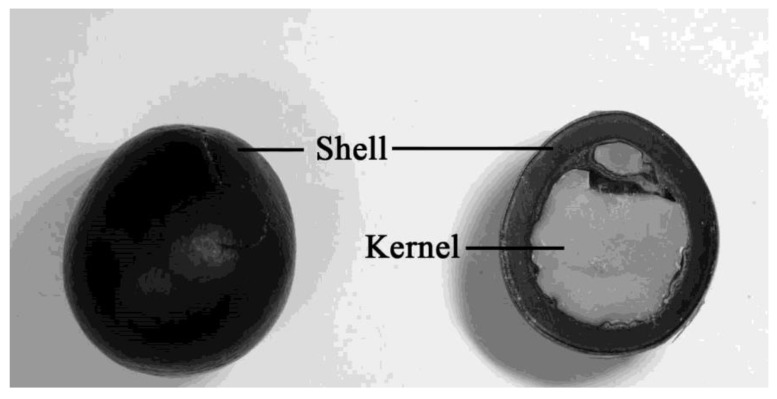
Pictures of the intact seed, seed shell and the kernel of the *Sapindus mukorossi* seed.

**Figure 2 ijms-20-02579-f002:**
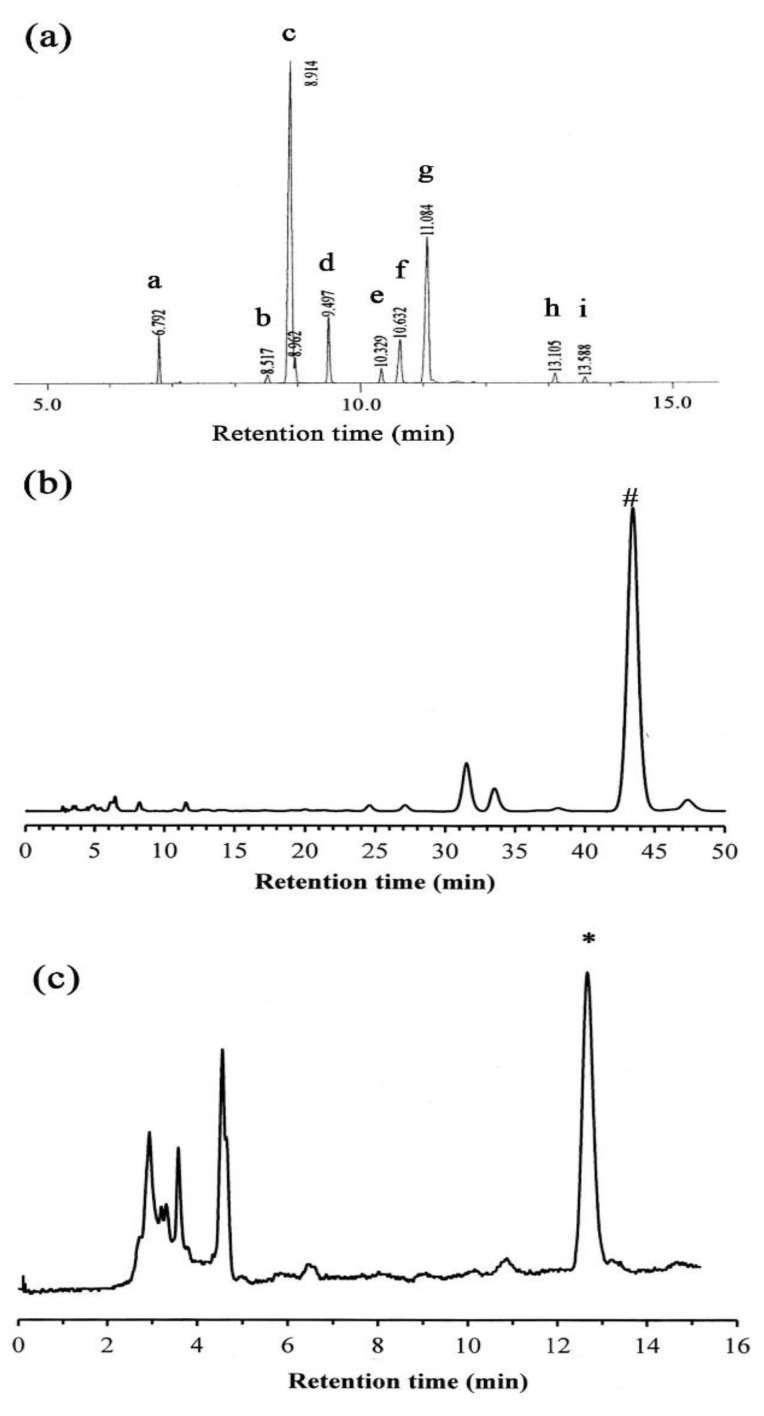
(**a**) Total ion chromatograms of *S. mukorossi* seed oil tested in this study. (**b**) δ-tocopherol and (**c**) β-sitosterol fractions from high-pressure liquid chromatography. # and * indicated the detected peaks of δ-tocopherol and β-sitosterol, respectively.

**Figure 3 ijms-20-02579-f003:**
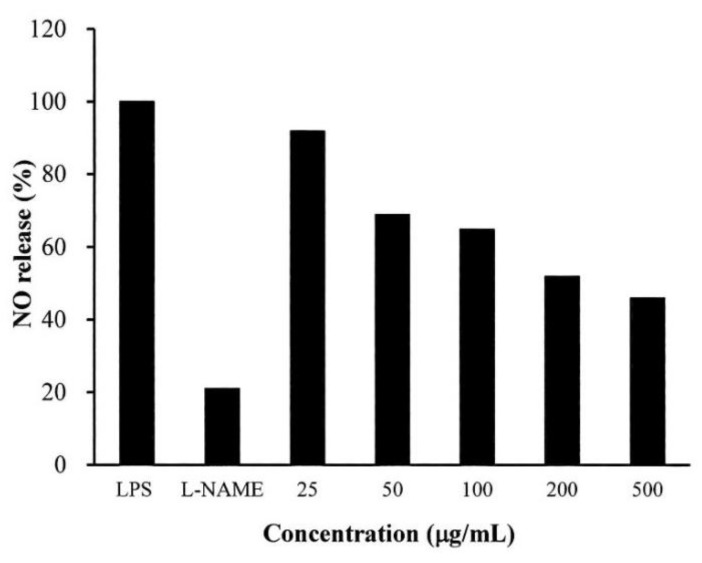
Nitric oxide (NO) release from the lipopolysaccharide (LPS)-treated RAW 264.7 cells decreased when the cells were pretreated with *S. mukorossi* seed oil. Data are from four independent experiments. The mean value of each group was normalized with LPS-only sample.

**Figure 4 ijms-20-02579-f004:**
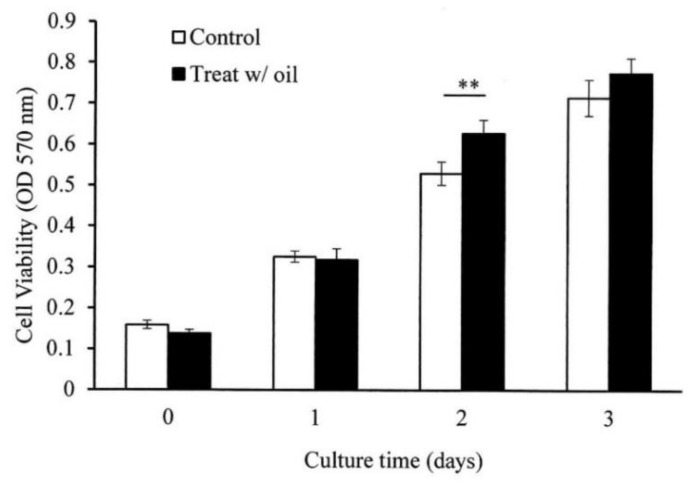
*S. mukorossi* seed oil significantly increased the viability of CCD-966SK cells by day 2. Data are presented as the mean ± SD. ** *p* < 0.01.

**Figure 5 ijms-20-02579-f005:**
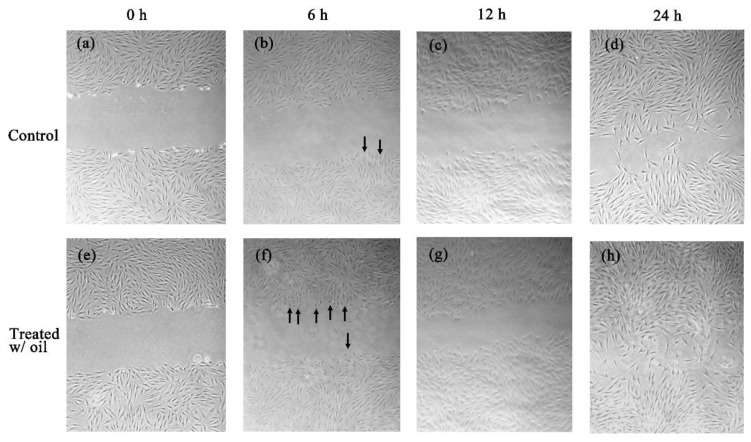
The control CCD-966SK cultured with *S. mukorossi* seed oil free medium at 0, 6, 12, and 24 h (**a**–**d**). *S. mukorossi* seed oil exhibits obvious effect on migration at 0, 6, 12, and 24 h (**e**–**h**) after incubation. The leading cells at the wound edge oriented toward the wound area 6 h after the scratch trauma was inflicted (black arrows).

**Figure 6 ijms-20-02579-f006:**
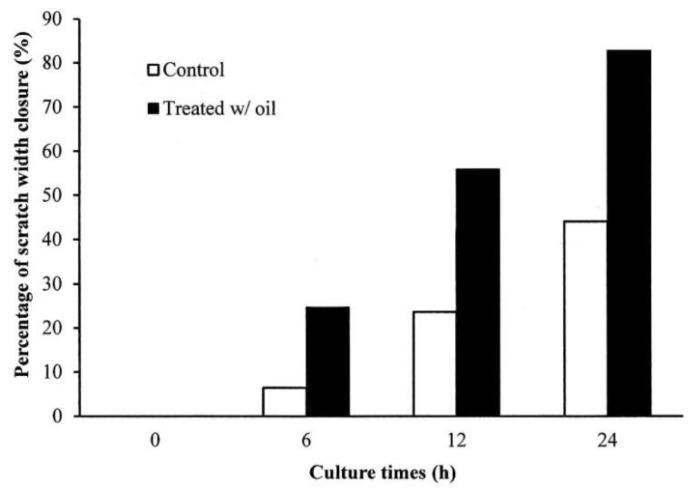
The percentage of scratch-width closure measured by quantifying the images of the scratch assay at 0, 6, 12, and 24 h after incubation. Data are from four independent experiments. The mean value of each group was normalized with LPS-only sample.

**Figure 7 ijms-20-02579-f007:**
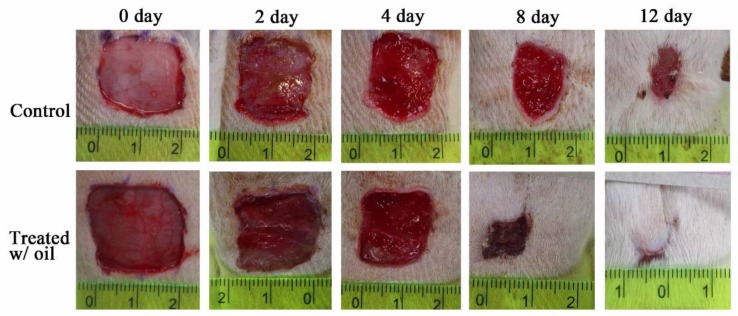
Photomicrographs of the wounds in rats after topical treatment with and without *S. mukorossi* seed oil on days 0, 2, 4, 6, 8, 10, and 12.

**Figure 8 ijms-20-02579-f008:**
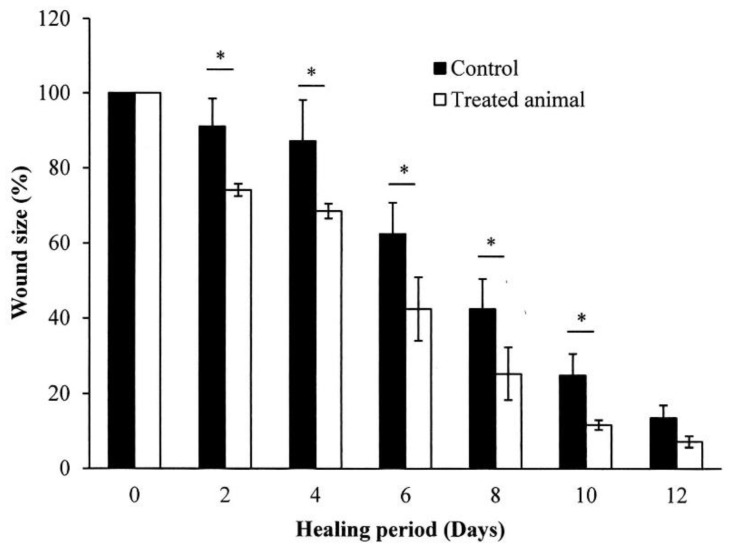
Quantification of the wound size in the rats treated with and without *S. mukorossi* seed oil. Data are presented as the mean ± SD. * *p* < 0.05.

**Table 1 ijms-20-02579-t001:** Fatty acid composition of *Sapindus mukorossi* seed-extracted oil by gas chromatography-mass spectrometry.

Peak	Retention Time (min)	Percentage	Fatty Acid
a	6.792	5.35	Palmitic acid (16:0)
b	8.317	0.90	Stearic acid (18:0)
c	8.914	52.46	Oleic acid (18:1)
d	9.497	7.19	Linoleic acid (18:2)
e	10.329	1.61	Linolenic acid (18:3)
f	10.632	6.84	Arachidic acid (20:0)
g	11.084	23.71	Eicosenic acid (20:1)
h	13.105	1.24	Behenic acid (22:0)
i	13.588	0.68	Erucic acid (22:1)

**Table 2 ijms-20-02579-t002:** Linearity of the standard curves of δ-tocopherol and β-sitosterol.

Compound	Calibration Equation ^a^	Retention Time (t_r_)	Correlation Coefficient (r^2^)
δ-Tocopherol	*Y* = 4903.9*X* ± 27882	12.67	0.9908
β-Sitosterol	*Y* = 3480.3*X* ± 7887.6	43.40	0.9996

^a^ The variable *X* is the concentration of the standard (mg/mL), and the variable *Y* is the peak area.

**Table 3 ijms-20-02579-t003:** Antimicrobial activity of *Sapindus mukorossi* seed oil extract.

Microorganism	Inactivation Rate ^a^ (%)
*Propionibacterium acnes*	>99.99
*Staphylococcus aureus*	>99.99
*Candida albicans*	99.9

^a^ The inactivation rate (%) = [1 – (CFU_sample_/CFU_control_)] × 100.

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
