# Peer review of "Effects of Sapindus mukorossi Seed Oil on Skin Wound Healing: In Vivo and in Vitro Testing"

_ijms, 2019, doi:10.3390/ijms20102579_

Round 1

Reviewer 1 Report

This manuscript by Chen and colleagues is well written and takes an interesting pharmacological look at the effects of S.mukorossi seed oil. Whilst I would like a more thorough investigation to understand how topical application of S.mukorossi seed oil affects wound healing in vivo (eg effects on inflammatory markers, endothelialisation, re-epithelialisation, fibroblast, granulation tissue, scar reduction etc) I do not believe this is the main aim of the manuscript. However I have a number of comments that need to be addressed.

1.       Is there a better quality picture for figure 1?

2.       There are no n numbers or error bars on figure 3 and figure 6. Generally all figures with graphs should also mention n numbers and statistics where relevant in the figure legend.

3.       In figure 5, it would be beneficial for the reader if cells which have oriented to the wound area (as stated in 2.5 cell proliferation) where circled/highlighted in some way

4.       Figure 8, Y-axis should say “wound size”

5.       How are wounds “scored” (pg 6, line 153) in the in vivo wound healing experiment, please clarify

6.       Line 194 should be “series” not serious

7.       Line 239 “plays as in ROS scavenging” should this be “plays a role in ROS scavenging”

8.       Please outline “saponification reaction” (line 277), are you letting the mixture stand for 15 minutes, please be more specific

9.       given that the bandages used to wrap the wounds are porous (line 382) and may soak up the oil, please discuss if this would affect the results and how to prevent the oil soaking into the bandage

Author Response

Reviewer: 1

This manuscript by Chen and colleagues is well written and takes an interesting pharmacological look at the effects of S.mukorossi seed oil. Whilst I would like a more thorough investigation to understand how topical application of S.mukorossi seed oil affects wound healing in vivo (eg effects on inflammatory markers, endothelialisation, re-epithelialisation, fibroblast, granulation tissue, scar reduction etc) I do not believe this is the main aim of the manuscript. However I have a number of comments that need to be addressed.

Author response: We thank the reviewer for all the comments that is an important aspect for the next step investigation.

1.       Is there a better quality picture for figure 1?

Author response: In the revised manuscript, the Figure 1 and 2 was replaced with new pictures with better quality.

2.      There are no n numbers or error bars on figure 3 and figure 6. Generally all figures with graphs should also mention n numbers and statistics where relevant in the figure legend.

Author response: The following statement was added to the legend of Fig. 3 and 6: “The data are from 4 independent experiments. The mean value of each group was normalized with LPS-only sample.” Since normalized data was present, no statistical analysis was performed.

3.          In figure 5, it would be beneficial for the reader if cells which have oriented to the wound area (as stated in 2.5 cell proliferation) where circled/highlighted in some way.

Author response: In Fig. 5, cells oriented to the wound area were identified with arrows.

4.      Figure 8, Y-axis should say “wound size”

Author response: In Figure 8, the typo error on Y-axis was revised to “size”.

5.      How are wounds “scored” (pg 6, line 153) in the in vivo wound healing experiment, please clarify

Author response: We thank the reviewer to point out this typo error, the word “scored” was revised to “obviously”.

6.      Line 194 should be “series” not serious.

Author response: We thank the reviewer to point out this typo error, the word “serious” was revised to “series”.

7.       Line 239 “plays as in ROS scavenging” should this be “plays a role in ROS scavenging”

Author response: We thank the reviewer to point out this writing error, the sentence was revised to “plays a role in ROS scavenging”.

8.      Please outline “saponification reaction” (line 277), are you letting the mixture stand for 15 minutes, please be more specific.

Author response: To describe the process more detain, the sentence was revised to “Then, for saponification, the mix was maintained at 80°C for 15 minutes.”

9.       given that the bandages used to wrap the wounds are porous (line 382) and may soak up the oil, please discuss if this would affect the results and how to prevent the oil soaking into the bandage

Author response: The CMC/HA/SA hydrogel [41,42] was prepared on “one side” of a non-woven fabric for releasing the extracted S. mukorossi seed oil. Thus the bandages did not contact the hydrogel which contain oil. We added a more detail explanation on line 377.

Reviewer 2 Report

The authors are invited to specify that the wound healing activity tested is onlyfor SKIN wound healing.

The conclusions enfatized the results, the authors areinvited to explain the real situation outlining the concentrations used and the outcomes recorded.

If the aim is to use a waste as source of bioactive compounds, could be useful describe the cost benefit of this strategy, directly consiquents of its large amount available in Japan.

In the selection of the cell lines used for the in vitro test, must be indicated why the authors preferred fibriblasts instead of keratinocytes.

Furthermore, the in vitro scratch test showed an evident result only after 24h of incubation but none explaination is reported, please furnish it.

Author Response

Reviewer: 2

1.     The authors are invited to specify that the wound healing activity tested is only for SKIN wound healing.

Author response: The phrase “wound healing” was revised to “skin wound healing” throughout the manuscript.

2.     The conclusions enfatized the results, the authors are invited to explain the real situation outlining the concentrations used and the outcomes recorded.

Author response: According to the reviewer’s comment, We added the following statement to the Conclusion: “The total amount of δ-tocopherol and β-sitosterol in the S. mukorossi seed oil was 73.9 ± 23.6 μg/mL and 232.64 ± 4.5 μg/mL, respectively. The inhibition effect of 1% S. mukorossi seed oil on P. acnes, S. aureus, and C. albicans are 99%. Addition of 500 μg/mL S. mukorossi seed oil results in a reduction of NO release by the lipopolysaccharide (LPS)-treated RAW 264.7 cells to 46%. The wound size of the oil-treated rats decreased significantly to almost 50% when S. mukorossi seed oil rich membrane was used as a dressing material.”

3.       If the aim is to use a waste as source of bioactive compounds, could be useful describe the cost benefit of this strategy, directly consequents of its large amount available in Japan.

Author response: According to the comment of the reviewer, we added the following statement on the last sentence of Conclusion: “Since the use of S. mukorossi seed oil involves a strategy of using a waste as source of bioactive compounds, it can conduct a cost benefit for skin care applications.”

4.       In the selection of the cell lines used for the in vitro test, must be indicated why the authors preferred fibriblasts instead of keratinocytes.

Author response: The reason for using fibroblast in this study was mentioned on line 253-255 as “These effects construct an ideal environment for fibroblast activity and collagen reorganization, with resulting acceleration of wound healing [14.37]. Since fibroblast proliferation and migration are vital steps in skin wound healing [12]”. In addition, the in vitro study using fibroblast was according to a previous wound healing investigation. We added this on line 338 as following “a cell viability assay was performed according to a previous study [36]”.  

5.     Furthermore, the in vitro scratch test showed an evident result only after 24h of incubation but none explaination is reported, please furnish it.

Author response: Actually the wound closure rate for the oil-treated cells is higher than that of the control cells at 6-hour. We added this time point on line 134-135.